# Serum Metabolic Profiles Based on Nuclear Magnetic Resonance Spectroscopy among Patients with Deep Vein Thrombosis and Healthy Controls

**DOI:** 10.3390/metabo11120874

**Published:** 2021-12-16

**Authors:** Melissa Quintero Escobar, Ljubica Tasic, Tassia Brena Barroso Carneiro da Costa, Danijela Stanisic, Silmara Montalvão, Stephany Huber, Joyce Maria Annichino-Bizzacchi

**Affiliations:** 1Hematology and Hemotherapy Center, Hemocentro, University of Campinas (UNICAMP), Campinas 13083-878, SP, Brazil; silmara@unicamp.br (S.M.); huber@unicamp.br (S.H.); 2Chemical Biology Laboratory, Department of Organic Chemistry, Institute of Chemistry, University of Campinas (UNICAMP), Campinas 13083-970, SP, Brazil; tbbcc@unicamp.br (T.B.B.C.d.C.); dacici.stanisic@gmail.com (D.S.)

**Keywords:** haemostasia, thrombosis, bleeding disorders, metabolomics, NMR

## Abstract

Deep venous thrombosis (DVT) is associated with significant morbidity and mortality. Studies on changes in the level of metabolites could have the potential to reveal biomarkers that can assist in the early detection, diagnosis, monitoring of DVT progression, response to treatment, or recurrence of DVT. In this scenario, the metabolomic analysis can provide a better understanding of the biochemical dysregulations of thrombosis. Using an untargeted metabolomic approach through magnetic resonance spectroscopy and multi- and univariate statistical analysis, we compared 40 patients with previous venous thrombosis and 40 healthy individuals, and we showed important serum differences between patients and controls, especially in the spectral regions that correspond to glucose, lipids, unsaturated lipids, and glycoprotein A. Considering the groups depending on risk factors and the local of the previous episode (lower limbs or cerebral system), we also noticed differences in metabolites linked to lipids and lactate. Comparative analyses pointed to altered ratios of glucose/lactate and branched-chain amino acids (BCAAs)/alanine, which might be associated with the fingerprints of thrombosis. Although samples for metabolomic analysis were collected months after the acute episode, these results highlighted that, alterations can still remain and may contribute to a better understanding of the complications of the disease.

## 1. Introduction

Deep venous thrombosis (DVT) is a common disease associated with significant morbidity and mortality, and represents the third most common cause of death, after cardiovascular disease and stroke [1,2]. Although most events evolve without complications, DVT can be complicated with a life-threatening pulmonary embolism (PE) [3], and approximately one-third of patients could develop moderate-to-severe post-thrombotic syndrome (PTS) [4]. The veins of the lower limbs are among the most commonly affected, but other sites can be compromised as well [5]. Cerebral venous thrombosis (CVT), although not very prevalent, usually occurs in young adults, especially in women under estrogen hormonal therapy, with a mortality around 10%, and up to 80% of patients recover with no neurological deficit [6].

DVT is a multifactorial disease and there are several risk factors that contribute to its development, including acquired and genetic factors [6,7]. Approximately 10% of individuals who have a first unprovoked episode of DVT will present a recurrent event in the first year, after anticoagulation withdrawal [4]. Even those patients with provoked DVT showed a high risk of recurrence, as demonstrated in studies of extended anticoagulation [8]. Recent guidelines indicate definitive anticoagulation for male patients with unprovoked DVT or for those diagnosed with thrombotic antiphospholipid syndrome (APS) [9,10]. However, in other situations, there is no clear definition regarding whether anticoagulation should be interrupted or not. This decision must consider the risk–benefits of anticoagulation, bleeding, and DVT recurrence. Thus, it is very important to discover the molecules and pathways involved in the process, and although there is an increasing number of research publications on DVT, these fields remain unexplored.

In this sense, the search for biomarkers that could identify patients with a higher risk of recurrence is a challenge. Higher levels of D-dimer are considered a marker of an increased risk of recurrence, and this measurement is included in Vienna, DASH, and Herdoo2 scores [11]. On the other hand, increased circulating metabolites, such as hyperhomocysteinemia, were thought to be part of the pathophysiologic link between B-group vitamins and venous thrombosis, but this remains controversial as well [12]. Emerging technologies, such as metabolomics, allow for an untargeted and thus unbiased analysis of the molecular underpinnings of the disease [13], leading to a better understanding of the disease’s pathophysiology. Metabolomics seek an analytical description of complex biological samples, aiming to characterize and quantify all the small molecules measurable in such samples [14]. Metabonomics broadly aim to measure the global, dynamic multiparametric metabolic response of living systems to biological stimuli or genetic modifications [15]. In practice, the terms metabolomics and metabonomics are often used interchangeably, and the analytical and modelling procedures are the same [14]. Here, we refer to both as metabolomics. This omics discipline enables the identification and quantification of the end products of cellular metabolism and allows us to understand the systemic changes in the complex multicellular systems.

Magnetic resonance (NMR) spectroscopy, in an untargeted approach, provides information on metabolites in a given sample [16]; however, for DVT, research is poorly explored [17]. Preliminary studies in animal models by NMR showed metabolic dysregulations on energy metabolism, sphingolipid, and adenosine metabolism [18]. Changes in metabolic pathways, including carbohydrates’, lipids’, and amino acids’ metabolism have been reported to be affected immediately after thrombosis [19]. Studies using mass spectrometry (MS) showed decreased acylcarnitines in the blood samples of patients with DVT and in those with a high risk of PE [18,19]. These findings suggest that carnitine metabolism may be dysregulated in venous thromboembolism (VTE), and provide evidence that metabolomic platforms might have the potential to be used to stratify the risks of patients with this condition [15]. An early and correct diagnosis of DVT is necessary for a pertinent treatment decision and a better result. The stages of clot maturation and resolution are not completely elucidated and defined, and there is no accurate biomarker of clot chronicity. The rise of VTE incidence, especially in the elderly, necessitates new diagnostic and prognostic biomarkers. However, it is also necessary to clarify the collecting time of biological samples to understand when the metabolites appear and when they become indicative of a diagnosis [20].

This work aimed to evaluate patients retrospectively, using a metabolomic point of view to find metabolic alterations, even months after an acute DVT episode. 

## 2. Results

### 2.1. Characteristics of Study Population

Among the 40 patients with a mean age of 38 years (18–78) at the time of thrombosis and enrolled in this study, 20 of them presented DVT in the lower limbs (called the DVT group, with a mean age of 45 years (20–78), 7 males, and 13 females), and 20 patients in the cerebral region (called the CVT group, with a mean age of 30 years (18–52), 1 male, and 19 females). Following the three years of observation, one patient presented PE and 12 were diagnosed with PTS in the DVT group. None of the patients in the CVT group presented recurrence. For detailed characteristics of the groups, see Appendix A
Table A1 and Table A2. All serum samples were compared with the respective control group. Table 1; Table 2 display the demographic and clinical variables of 32 women and 8 men, fulfilling the inclusion criteria. The median time between the thrombotic episode and blood collection was 13 months, and six patients were under anticoagulation with warfarin (*n* = 5) or rivaroxaban (*n* = 1) at the moment of blood collection. For the time between blood collection and the thrombotic episode, see Appendix A
Table A1 and Table A2. Analysis by age and sex between the thrombosis and control groups showed no significant differences.

The classification of patients according to BMI [21] showed that 20% of the patients presented normal weights (mean 21.75 kg/m^2^), 35% were overweight (mean 27.60 kg/m^2^), 15% presented obesity class I (mean 32.50 kg/m^2^), 10% obesity class II (mean 36.90 kg/m^2^), 5% morbid obesity (mean 51.15 kg/m^2^), and in 15%, BMI information was not available (Table 2). In the control group, ≈48% were normal weight (mean 22.18 kg/m^2^), 30% overweight (mean 27.79 kg/m^2^), 15% classified as obesity class I (mean 32.72 kg/m^2^), ≈3% as obesity class II (mean 35.75 kg/m^2^), and 5% as non-reported BMI (Table 3). The analysis of thrombosis versus control group based on BMI showed a significant difference (*p* = 0.038).

### 2.2. Metabolomic Data Analysis

#### 2.2.1. ^1^H-NMR Data of Serum Samples in Thrombotic Patients vs. Controls

A total of 15 known metabolites were identified using ^1^H-NMR CPMG data (Figure 1A). Different groups of lipids, amino acids such as alanine, isoleucine, leucine, valine, glutamine, aromatic amino acids—tyrosine, histidine, and phenylalanine—organic acids, and glucose were identified. For the chemical shifts’ assignments, see Table A3. Although BCAAs have not yet been reported to be associated with venous thrombosis, in this study, we showed alteration in leucine, valine, alanine, and lipoproteins in the thrombosis group (Figure A1).

#### 2.2.2. Multivariate Statistical Analysis

A comparative analysis, based on orthogonal partial least squares discriminatory analysis (OPLS-DA), showed a sample clustering related to patients or controls (Figure 1B) with R^2^X, R^2^Y, and Q^2^ values of 3%, 23%, and 7%, respectively. The thrombosis group was well dispersed, maybe because of the different groups of analyzed samples, and the use of a constructed model alone was not enough to consider the model predictable. However, it was possible to identify some variables according to the value of their PLS regression coefficients, classifying them as the relative importance of variables in projection or VIP. According to VIP scores values >2.0, lipids (δ 1.34; 2.04), alanine (δ 1.47), valine (δ 1.04), glutamine (δ 2.44), and glucose (δ 3.23; δ 3.55; δ 3.83; δ 3.73) were responsible for class separation in the thrombotic group, and all presented significant differences by the U test (Wilcoxon–Mann–Whitney, *p* < 0.05) (Figure 1C). Additionally, glucose and lipids that are exclusively represented by unsaturated lipid function and/or glycoprotein A also presented *p* < 0.05, according to the linear regression model (Table A4).

The receiver operating characteristic (ROC) curves of the metabolites discussed pointed to an acceptable capacity to discriminate patients (Figure A2), with the greatest area under the ROC curve (AUC = 0.73) values observed for glutamate and valine.

#### 2.2.3. Risk Factor Analysis

It is already known that thrombosis can be triggered by a known risk factor, called provoked thrombosis. In our cohort of patients, hormonal contraceptives (OAC) and obesity were the principally acquired known risk factors, especially OAC for CVT (75%) and obesity for DVT (35%) (Table 1). The BMI showed significant differences between the groups (Table 1), so we also explored the intrinsic differences related to the condition of the BMI between the groups (Figure 2A). Although this PLS-DA model did not represent a clear separation between classes, lipids (δ 0.89 to δ 1.31) and lactate (δ 1.32) alteration were found for overweight and obese class I patients (Figure 2B). In the absence of a risk factor, thrombosis is considered unprovoked, and in this situation, an unknown prothrombotic state may occur. To investigate if metabolic alterations could be observed in unprovoked DVT, we also performed a multivariate analysis. PLS-DA-based profiling showed a clustering tendency (Figure 2C). Once more, alterations in lipids (δ 0.88–1.31) were identified as VIP for patients with unprovoked DVT. It was possible to associate one chemical shift (δ 1.33) to the provoked condition, which corresponds to lactate (Figure 2D).

#### 2.2.4. Subgroups of Patients

As mentioned previously, the multivariate analysis showed the dispersion of the thrombosis group (Figure 1B), and this could be related to the different types of samples that we analyzed (Table 1). This led us to separate the thrombosis group into subgroups according to the site of venous occlusion—DVT for lower limbs or CVT for the cerebral system—and to compare them by multivariate analysis of the NMR data with their respective controls. Among the patients, 20 presented DVT with a mean age of 45 years (20–78), 7 males, and 13 females, and 20 patients presented CVT—with a mean age of 30 years (18–52), 1 male, and 19 females. The control groups were specific for each subgroup, 20 healthy subjects, called the DVT control group—with a mean age of 47 years (26–66), 7 males, and 13 females—and 20 healthy subjects, called CVT control group—with a mean age of 33 years (23–55), 3 males, and 17 females.

This analysis showed a sample clustering for DVT and CVT vs. controls (Figure 3A,B, respectively). The VIP created from the models’ showed alterations in glucose and glutamine for the DVT group (Figure 3A), and in glucose, alanine, and lipids for the CVT group (Figure 3B). When comparing the DVT subgroups against the CVT group, all metabolites mentioned were also found to be important, except for alanine (Figure 3C). The VIP scores for each created model are shown in Table A4.

## 3. Discussion

Research efforts are currently seeking alternative blood biomarkers that are capable of improving patient diagnosis, treatment, and stratification, identifying those patients that may benefit from different treatment schedules, and particularly in thrombosis, extended anticoagulation [18]. In this study, using a metabolomic approach, we observed global differences among the serum metabolic patterns of thrombotic patients and controls. As seen in Figure 1, discriminatory metabolites such as lipids, glucose, alanine, glutamine, and valine were identified, and showed significant differences between patients and controls. The area-under-the-curve (AUC) values of the receiver operating characteristic (ROC) curves were compared for every single individual metabolite. Glutamate and valine were the most influential metabolites in discriminating between the thrombotic and control groups. Even though this is not a study with a diagnostic objective, and as there is no predictive model or independent cohort testing in this study, we indicate that this is one of the caveats of this study.

Common variables, called confounders, such as age, sex, lifestyle, BMI, estrogen use, and others, might explain the alterations of lipids in thrombotic patients [22]. However, it is important to note that significant differences between patients and controls were observed only regarding BMI and estrogen use. When we explored the intrinsic differences related to the BMI condition between the groups, alterations in lipids (from δ 0.89 to δ 1.31) and lactate (at δ 1.32) were observed in overweight and obesity I patients.

As the blood samples were collected months after the thrombotic episode, we can only speculate that if lipid alteration was present before the thrombosis, it could interfere with hemostasis and inflammation [23]. This can be demonstrated by changes in levels of hemostatic factors [24] and C-reactive proteins (CRP) [25]. Otherwise, even if these alterations occurred a long time after the venous episode, they differentiate patients and controls. The role of these alterations for DVT complications, such as recurrence, can be investigated in a study with longer follow-ups and with a higher number of patients.

Lactate was also found to be important for the provoked thrombosis. This alteration was previously reported as caused by a number of conditions, including shock, cardiac arrest, trauma, ischemia, diabetic ketoacidosis, genetic disorders, and other conditions, which is an indicator of a non-specific factor unless thoughtfully coupled with the overall clinical picture [26]. Maekawa et al. identified lactate as one of the blood metabolites reflecting fresh venous thrombus in a jugular model of venous thrombosis in 2019, probably as a result of active glycolysis of thrombus cellular components, predominantly erythrocytes [27]. As our samples were not obtained during the acute phase, and were obtained even months after the thrombotic episode, these results nonetheless demonstrated that lactate alterations appear to remain for a long time.

As mentioned above, in our cohort of patients, hormonal contraceptives and obesity were the principal acquired risk factors. Normally, obesity interacts with other environmental or genetic factors, increasing the risk of thrombosis [28]. Particularly, central and abdominal obesity is associated with increased thrombin formation and decreased fibrinolysis [29]. On the other hand, one of the common risk factors for thrombosis in women is the use of hormonal contraceptives. The contraceptives showed a strong association with thrombosis due to their estrogenic component, as estrogen increases the levels of coagulation factors and decreases the levels of anticoagulant proteins such as protein S [29,30]. Hormonal contraceptive use increases the risk of CVT in women of reproductive age seven-fold when compared to those not using this method of contraception [31]. Of the 19 patients that presented CVT, 75% were using hormonal contraceptives, emphasizing the importance of this risk factor in this population. However, when blood samples for metabolomic analysis were collected, hormonal use was already discontinued.

A comparison between thrombotic patients and controls showed differences in blood serum lipids (fatty acyls, especially, in very-low-density lipoproteins, VLDL) (Figure 1C), as well in the DVT subgroup (Figure 3C). There is evidence that VLDL and remnant lipoproteins induce a procoagulant state. These lipoproteins have the capacity to activate platelets and the coagulation pathway, and to assemble the prothrombinase complex. There are differences in the capacity of lipoproteins to generate thrombin, and this effect is further altered for low-density lipoproteins (LDL), in particular by the oxidation of the unsaturated fatty acids [32]. While platelets have the largest capacity to generate thrombin via the prothrombinase complex pathway, VLDL has a much higher capacity than oxidized LDL. VLDL compared to oxidized LDL has the ability to increase thrombin formation by approximately 10-fold [33]. Once an individual presents thrombosis, the predisposition probably remains, mediated by the interaction between triglyceride-rich lipoproteins, platelets, and the vascular endothelium. Since our study is retrospective, we can speculate that these alterations could contribute as a causal factor or as a consequence of the thrombotic episode, and could participate in future complications. We have to point out that the altered profile still remains, even after the statistical correction of BMI, sex, and age.

Other metabolites’ alterations were identified and could be linked to energy or amino acids’ metabolism pathways. Alterations in glucose are associated with the VTE as a risk factor, given that hyperglycemia can activate coagulation [27]. However, here we observed that glucose decreased in the thrombotic group and the DVT subgroup when compared with their controls, and surprisingly, increased in the CVT group. However, it is already known that hyperglycemia was reported as a strong predictor of poor clinical outcomes in patients with CVT [34]. We also found alterations in the ^1^H-NMR spectral region that correspond to glycoprotein A (GlycA). Recently, glycoprotein profiles by NMR have emerged as a potential biomarker that reflects systemic inflammation in type 2 diabetes, obesity, cardiovascular events, and other pathological processes [35,36,37,38,39,40]. Levels of GlycA show an abundance of mobile *N*-acetyl sugar groups present on glycoproteins in circulating blood, which are involved in the acute phase response of different inflammatory disease states [40]. This inflammatory biomarker could also be associated with thrombotic events and could be used as an analytic or clinical tool that may complement or provide advantages over existing clinical markers of systemic inflammation. It is well understood that inflammation can activate the coagulation cascade by several mechanics: down-regulation of antithrombin and other anticoagulant mechanisms, tissue-factor-mediated thrombin generation, and impaired fibrinolysis. Thrombin itself is known to induce IL-6 and IL-8 in endothelial cells, which play a role in the maintenance of inflammation. Furthermore, the endothelial injury induced by pro-inflammatory cytokines worsens the coagulation balance [41]. Nevertheless, it will be interesting to analyze samples in the acute phase to confirm this finding, and to make additional studies to evaluate its role.

The branched-chain amino acids (BCAAs) leucine, isoleucine, and valine are important resources for energy production and regulators of metabolic and nutrient signals. BCAA catabolism is a regulator of platelet activation, and BCAA dysfunction could be associated with arterial thrombosis risk [42]. Although BCAAs have not yet been reported to be associated with venous thrombosis, in this study, we showed alterations in leucine, valine, alanine, and lipoproteins in the thrombosis group (Figure A1).

Previous studies reported that aspirin, an anti-aggregating drug, can reduce the risk of recurrence of DVT by 30% [43]. An elucidation of the multiple mechanisms involved in platelet induction of venous thrombosis provides opportunities to selectively inhibit this pathway that is relevant to the pathophysiology. Sung et al., 2018 proposed that thrombosis could have a perturbed turnover in the TCA cycle [18], and this could be indicated by metabolites such as valine, alanine, and glutamine that were also found to be decreased in thrombotic patients. These metabolites could be associated with the pathological condition, as previously reported [18].

### Strengths and Limitations 

Strengths of this study include a well characterized investigated cohort selected based on rigorous exclusion criteria—no DVT recurrence, APS, cancer, infection, renal, hepatic or inflammatory disease, as well as, the use of corticosteroids, and tobacco. The use of a homogenous cohort matched with a similar control group, and inclusion of a metabolomics analysis to highlight the alterations that remain after thrombotic episodes.

The limitations of this study are the relatively small number of patients having in mind such a common disease, the lack of inflammatory or thrombotic markers at the time the blood samples were collected for the metabolomic analysis, as well as the design of the retrospective study. 

Although the data obtained herein provided important features in patients with previous venous thromboembolic disease that distinguish them from controls, future studies are still needed to confirm the findings in patients with acute DVT.

Since the purpose of the current study was to identify metabolic alterations even months after acute DVT episode, the next step is a validation analysis including diverse populations and different disease subtypes.

## 4. Materials and Methods

### 4.1. Study Population

We studied 40 serum samples from patients with a previous history of thrombosis. The patients were assisted at the Outpatient Clinic of Hemocentro of Unicamp (Campinas, SP, Brazil) between 2015 and 2018, and were selected according to the inclusion criteria: aged 18 to 80 years, a unique previously confirmed DVT of the lower limbs or CVT up to three years after acute thrombotic episode. The exclusion criteria were DVT recurrence, APS, cancer, infection, renal, hepatic, or inflammatory disease, as well as the use of corticosteroids and tobacco. Forty healthy subjects from the same geographic area, and workers from Hemocentro were used as controls. The exclusion criteria for the healthy individuals were the same as for the patients, including previous DVT. The diagnosis of lower-limb DVT was established by Doppler ultrasound, and the diagnosis of CVT was confirmed by cerebral magnetic resonance or cerebral angiotomography. The project was approved by the local medical research review board and ethics committee (file number 18501718.9.0000.5404).

### 4.2. Blood Collection

Blood was collected by venipuncture after minimal stasis from the antecubital vein in standard citrated tubes (Vacutainer; Becton Dickinson, Franklin Lakes, NJ, USA), and centrifuged at 1372 *g* for 15 min at room temperature. The serum was stored in aliquots at −80 °C until analysis.

### 4.3. ^1^H-NMR Spectroscopy Analyses

For ^1^H-NMR spectra, 250 μL of each serum sample was mixed with 250 μL of deuterium oxide solvent (D_2_O, 99.9% with 0.03% of trimethylsilyl propanoic acid, TSP, from Sigma–Aldrich, Andover, MA, USA), and transferred into 5-mm NMR tubes. The ^1^H-NMR spectra were acquired using a Bruker AVANCE III 600 spectrometer (Bruker Biospin, Karlsruhe, Germany) at 25 °C operating at 600.13 MHz and equipped with a TBI (Triple Resonance Broadband Inverse) probe. Regular one-dimensional ^1^H-NMR spectra with a T_2_ filter was recorded using the pulse sequence CPMG (Carr–Purcell–Meiboom–Gill) with ns = 128.

### 4.4. Data Analysis: NMR Data Processing and Statistics

All serum ^1^H-NMR spectra were manually processed to correct their phases and baselines. Chemical shifts were referenced to the TSP signal (δ 0.00), using MestReNova software (9.0.1-13254). The spectra were divided into regions with an equal width of 0.005 ppm. The HDO (δ 4.70–5.55) was excluded from the analysis. Samples were normalized by a reference sample for Probability Quotient Normalization (PQN). The metabolites were assigned based on chemical shifts, coupling constants, 2D NMR spectral characteristics, and in concordance with the Human Metabolome Database (HMDB) and BioMagResBank (BMRB) databases [44,45]. MetaboAnalyst was used to develop ROC curves of individual metabolites.

### 4.5. Multivariate Data Analysis

The ^1^H-NMR binned data were used as a data matrix and the multivariate analysis was performed using MATLAB software (v R2015a, Mathworks Inc, Natick, MA, USA) and the MetaboAnalyst platform [46]. The matrices were constructed with 80 or 40 samples against 1628 variables (bins) for ^1^H-NMR CPMG, and for visualization data, we used pareto scaling (mean-centered and divided by the square root of the standard deviation of each variable). A principal component analysis (PCA) was performed for all samples to explore inherent groupings in the data and identify outliers. Data were modelled with the supervised method of orthogonal partial least squares discriminant analysis (OPLS-DA) to find the metabolite differences between the groups. Variable importance in projection (VIP) scores were estimated according to the PLS-DA model [47].

Multivariate analysis was also performed using linear regression models, with the values of the ^1^H-NMR bin data as dependent variables and, as independent variables, the subgroups of patients (by type of thrombosis) and the adjustment variables (sex, age, and body mass index—BMI). The *p*-values < 0.05 were considered statistically significant. All analyses were performed using the R version 3.6.1 (5 July 2019), copyright (C) 2019, the R Foundation for Statistical Computing.

## 5. Conclusions

There is a great need for low-cost, rapid, simple, and reliable methods that can be used as important point-of-care methods to evaluate the gravity and monitor common diseases’ progress such as venous thrombosis. To this end, the identification of biomarkers has appeal, and given the complex nature of thrombosis, it has been difficult to earmark potential ones. Here, we provided new insights into the importance of some serum metabolites, such as glucose, lactate, alanine, BCAAs, and lipids with important differences in VLDL fatty acid levels in thrombotic patients. Comparative analyses also pointed to altered ratios of glucose/lactate and BCAAs/alanine, which might be associated with the fingerprints of thrombosis, especially high VLDL for the DVT group, which might lead to a procoagulant state.

Although the data found are in agreement with similar studies, this is a retrospective study. This is, to our knowledge, one of the few articles carried out in patients with venous thrombosis. The results obtained can contribute to novel perspectives towards a better comprehension of the disease and might open up the possibility of including metabolomics by ^1^H-NMR as an important add-on to the clinical research.

## Figures and Tables

**Figure 1 metabolites-11-00874-f001:**
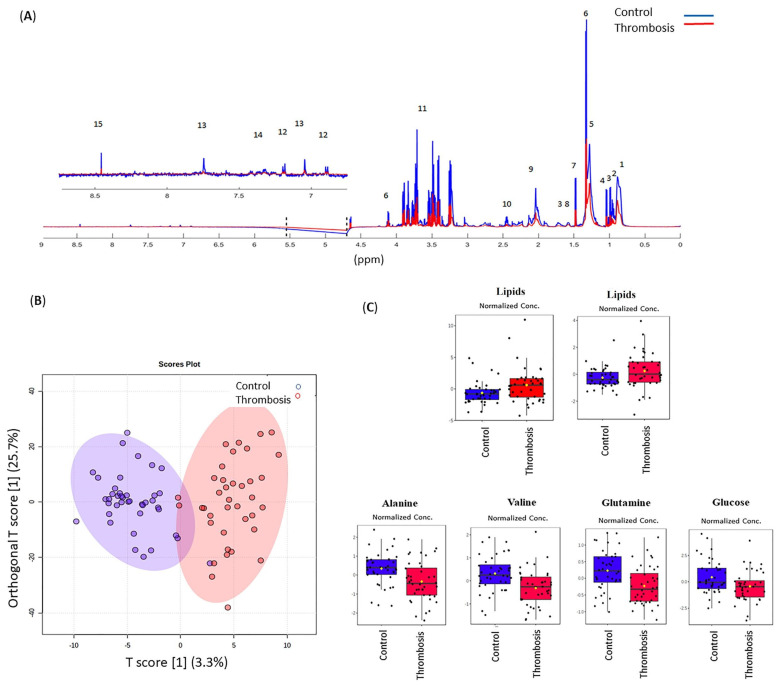
(**A**) ^1^H-NMR CPMG spectra illustrating differences between thrombosis and control group. The identified metabolites were: 1. Fatty acyl chain terminal methyl group hydrogens -CH_3_; 2. Isoleucine; 3. Leucine; 4. Valine; 5. Lipids CH_3_(CH_2_)n; 6. Lactate; 7. Alanine; 8. Fatty acyl chain hydrogens from the -CH_2_- group next to the carboxyl group, as in CH_2_CH_2_C(O); 9. Fatty acyl chain CH_2_CH=; 10. Glutamine; 11. Multiple glucose hydrogens; 12. Tyrosine; 13. Histidine; 14. Phenylalanine; 15. Formate. The HDO (δ 4.70–5.50) regions were removed before the analyses, indicated with the dotted lines. The inset shows the expanded (δ 6.50–8.50) spectral regions; (**B**) Multivariate OPLS-DA; (**C**) Relative metabolite levels (measured as peak intensities) by VIP score. The black dots represent the metabolite levels in all samples, and the yellow diamond represents the average value. Patients (Thrombosis) are illustrated in red, and controls in blue.

**Figure 2 metabolites-11-00874-f002:**
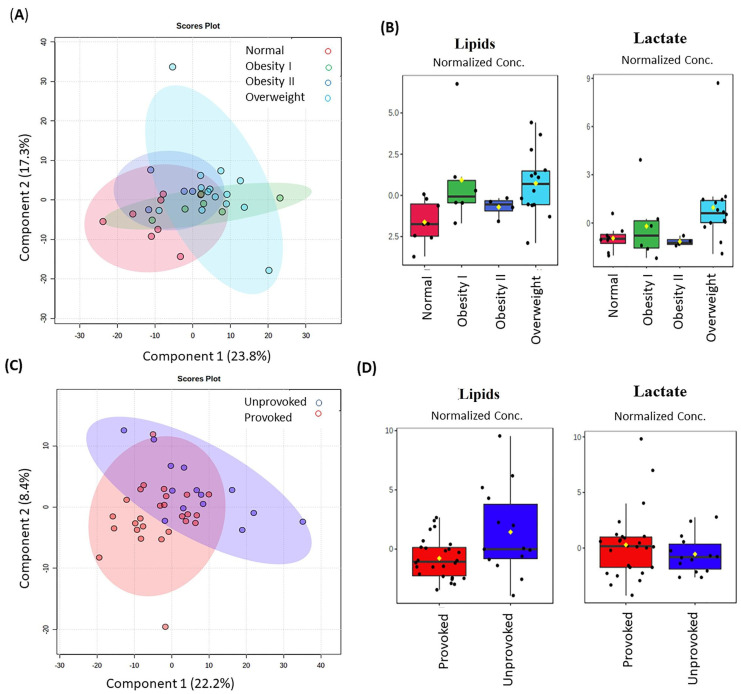
(**A**) PLS-DA according to the BMI (accuracy 53%, R^2^ 0.265, and Q^2^ -0.05) (**C**). PLS-DA between provoked or unprovoked known risks (accuracy 0.253, and Q^2^ 0.062) (**B**–**D**) Relative metabolite levels (shown as Normalized Conc. and measured as peak intensities) by VIP—lactate (δ 1.32–1.34) and lipids (δ 0.88–1.31) were responsible for class separation for obesity I and overweight groups, and for provoked and unprovoked groups, respectively. However, only spectral data at δ 0.90 presented statistically significant differences (*p* < 0.05) between the conditions. These regions correspond to lipids. The black dots represent the metabolite levels in all samples and the yellow diamond represents the average value. In red are thrombosis patients with normal BMI; in green, patients with obesity I; in dark blue, patients with obesity II; in light blue, overweight patients. Provoked are in red and unprovoked are in blue.

**Figure 3 metabolites-11-00874-f003:**
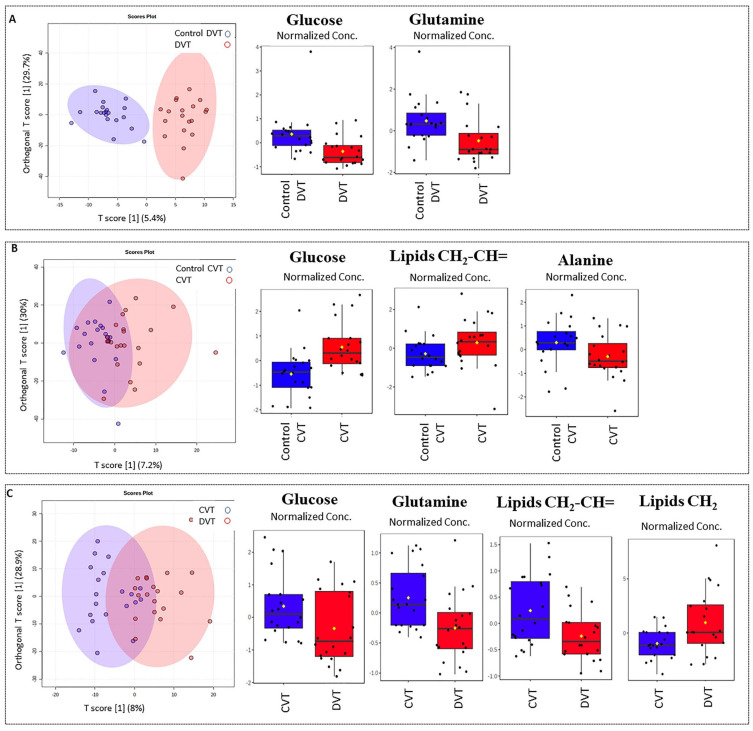
Illustration of results of OPLS-DA from ^1^H-NMR CPMG data showing subgroups with the respective VIP responsible for class separations. The levels of metabolites, shown as Normalized Conc., were expressed as peak intensities. (**A**) DVT vs. DVT’s control R^2^X 30%, R^2^Y 35%, and Q^2^ 13%; (**B**) CVT vs. CVT’s controls R^2^X 30%, R^2^Y 10%, and Q^2^ 12%; (**C**) DVT vs. CVT patients R^2^X 8%, R^2^Y 10%, and Q^2^ 19%.

**Table 1 metabolites-11-00874-t001:** Clinical summary of patients with different types of venous thrombosis. DVT: deep venous thrombosis in the lower limbs; CVT: cerebral veins thrombosis; BMI: body mass index.

Variable	A. Thrombotic Patients (*n* = 40)		B. Controls (*n* = 40)	*p*-Value
				A vs. B
Age, mean years	38 (18–78)		40 (23–66)	0.178
Male, *n* (%)	8 (20)		10 (25)	1.000
Body mass index, kg/m^2^	29 (19-60)		26 (18–36)	0.038
	DVT (*n* = 20)	CVT (*n* = 20)		0.045
Age, mean years	45 (20–78)	30 (18–52)		
Male, *n* (%)	7 (35)	1 (5)		
Body mass index, kg/m^2^	32 (20–60)	27 (19–38)		
Risk factors				
Provoked, *n* (%)	10 (50)	16 (80)		
Family history of VTE, *n* (%)	2 (12.5)	3 (16)		
Hormonal contraceptive	6 (30)	15 (75)		
Obesity	7 (35)	5 (25)		
Surgery	3 (15)	1 (5)		
Gestational/postpartum	1 (5)	1 (5)		
Anticoagulant use at the moment of blood collection				
Warfarin, *n* (%)	2 (10)	3 (15)		
Rivaroxaban, *n* (%)	1 (5)			

**Table 2 metabolites-11-00874-t002:** Characteristics of patients in relation to BMI.

Patients	Normal	Overweight	Obese Class I	Obese Class II	Morbid Obesity
BMI, mean (min–max), kg/m^2^	21.75 (19.27–24.80)	27.60 (25.78–29.8)	32.5 (30.40–34.10)	36.9 (35.86–37.60)	51.15 (42.70–59.60)
Participants, *n* (%)	8 (20)	14 (35)	6 (15)	4 (10)	2 (5)
Women, *n* (%)	8 (20)	9 (18)	5 (12.5)	4 (10)	2 (5)
Age, mean (min–max), years	31 (18–78)	40 (21–73)	39 (22–67)	35 (23–54)	53 (49–56)
DVT, *n* (%)	1 (2.5)	8 (20)	4 (10)	1 (2,5)	2 (5)
CVT, *n* (%)	7 (17.5)	6 (15)	2 (5)	3 (7.5)	

**Table 3 metabolites-11-00874-t003:** Characteristics of controls in relation to BMI.

Controls	Normal	Overweight	Obese Class I	Obese Class II
BMI, mean (min–max), kg/m^2^	22.18 (18.87–24.92)	27.79 (26.22–29.33)	32.72 (30.08–33.95)	35.75
Participants, *n* (%)	19 (47.5)	12 (30)	6 (15)	1 (2.5)
Women, *n* (%)	17 (42.5)	7 (17.5)	3 (7.5)	1 (2.5)
Age, mean (min–max), years	34	43 (23–66)	47 (33–57)	52

## Data Availability

The data presented in this study are available in article.

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
