# Peer review of "Serum Metabolic Profiles Based on Nuclear Magnetic Resonance Spectroscopy among Patients with Deep Vein Thrombosis and Healthy Controls"

_metabolites, 2021, doi:10.3390/metabo11120874_

Round 1

Reviewer 1 Report

Authors found alteration of serum metabolite levels in patients with previous deep vein thrombosis and cerebral venous thrombosis. 

Major comment
1. Study design. The study questions/clinical problems and aims are unclear. Is the study design appropriate for resolving the questions/problems? 

2. Did the authors have any data on blood/thrombotic/inflammatory makers, such as leukocyte/platelet counts, D-dimer and thrombin antithrombin complex. Please show relationship between them and each metabolite levels.

3. The serum metabolite levels reflect systemic and/or local metabolic states of the patient. In discussion, the authors focused only on thrombus formation and thrombus itself, and do not consider the major metabolic organs; liver, adipose tissue, skeletal muscle tissue (especially DVT patients) and brain (cerebral thrombosis patients).

Minor comments

1. 2.1. Table 1. Author did not describe a significant difference in BMI between the groups.

2. Table 2. Obesity type 1 and 2. Obesity class 1 and 2 are suitable.

3. All figures. The size of the letters in all figures are too small to see.

4. Appendix B figure 1 did not appear in the result section.

5. Table A1 and A2, age of the patients. For example, 30s or 40s rather than 31 or 45 are adequate as personal information.

Reviewer 2 Report

The manuscript identifies metabolic biomarkers among patients (serum samples) suffered from Deep Venous Thrombosis. The authors compared the serum samples from patients, who has suffered from DVT event, with healthy controls using 1D NMR based metabolomics approaches and identified BCAA, Glucose, amino acids, lipids among others as metabolic biomarkers for Venous thrombosis. The study may be interest to clinicians/researches studying this area.

The study is conducted with descent cohort size and authors considered other factors in their metabolomics analysis (e.g. Age, sex, BMI, disease status, etc.). I think this study can be published. I just have few questions/comments before consideration for publication,

  • The study is missing validation cohort. The findings can be further corroborated if authors have created models based on these differential features and tested on independent validation sets.
  • Authors should mention in methods section how did they quantify the metabolite levels from 1D peak intensities and did authors used any deconvolation methods in their analysis ?
  • Why did authors choose serum as choice of sample over plasma or blood ?
  • Are these patients known to have any VT at the time of blood draw ?

Comments:

The images resolution seems to be low please use HR images for better clarity.

Please review manuscript for typos.

Reviewer 3 Report

The study conducted by Escobar et al suggest a novel approach for the early detection of and diagnosis of DVT by monitoring the level of serum metabolites. The study suggests an alteration in the levels of serum metabolites in patients suffering from DVT in comparison to healthy controls. The potential drawback of the study is its retrospective design and small sample size, which has been rightly pointed out by the authors. Despite this, the manuscript has been informative and the study has been performed comprehensively, which merits its publication in the present form. 

Author Response

Thank you for the valuable feedback, we appreciate the opportunity to publish our findings.

Round 2

Reviewer 1 Report

The authors did not answer the major comment 1 and 3, and did not revise the manuscript. The study questions/clinical problems and aims remain unclear. I have concern on the study design.

Author Response

Responses to Reviewer 1

Comments and Suggestions for Authors

The authors did not answer the major comment 1 and 3, and did not revise the manuscript. The study questions/clinical problems and aims remain unclear. I have concern on the study design.

Answer: We understand the reviewer’s concerns. As mentioned in the previous comments this is a retrospective study and we refrased the objectives to be more clear. Also, it is important to have in mind that the cases in the studied cohort were selected based on rigorous exclusion criteria - no DVT recurrence, APS, cancer, infection, renal, hepatic or inflammatory disease, as well as, the use of corticosteroids, and tobacco.

  1. Study design. The study questions/clinical problems and aims are unclear. Is the study design appropriate for resolving the questions/problems?

Answer: We understand the reviewer’s concerns. As a retrospective study we did not intend to generate a predictive model of thrombosis based on the metabolomic profile data. However, we consider that with the data obtained in this study we provided important features in patients with previous venous thromboembolic disease that differentiate them from controls, with novel perspectives towards a better comprehension of this disease.

  1. The serum metabolite levels reflect systemic and/or local metabolic states of the patient. In discussion, the authors focused only on thrombus formation and thrombus itself, and do not consider the major metabolic organs; liver, adipose tissue, skeletal muscle tissue (especially DVT patients) and brain (cerebral thrombosis patients).

Answer: We measured creatinine levels and prothrombin time, and both groups showed normal results, suggesting no renal or hepatic dysfunction. It is important to point out that blood samples for metabolic analysis were collected months after the acute thrombotic episode, and did not reflect thrombus formation or thrombus itself.

Reviewer 2 Report

Thank you for authors comments to my queries. Few suggessions before consideration for publication,

Authors should include ROC curves and describe it in main body or in supplimental. As there is no predictive model or independent cohort testing in this study authors should indicate this as one of the caveat in the dicussion.

Given there is no quantification carried out in this study, in manuscript either word "levels" should be used instead of "concentration" or "concentration (measures as peak intensities)" should be used wherever appropriate.

Author Response

Responses to Reviewer 2

Comments and Suggestions for Authors

Authors should include ROC curves and describe it in the main body or in supplemental. As there is no predictive model or independent cohort testing in this study authors should indicate this as one of the caveats in the discussion.

Answer: Thank you for the suggestions. We understand the basis for the reviewer’s concern and we added to the manuscript in appendix B section Figure 2, and explained in more detail the studied cohort.

Given there is no quantification carried out in this study, in manuscript either word "levels" should be used instead of "concentration" or "concentration (measures as peak intensities)" should be used wherever appropriate

Answer: Thank you for the observations. We have added the new information to the figures, and the explanation on how metabolite levels were measured